# Dr. Docker: A Large-Scale Security Measurement of Docker Image Ecosystem

## Abstract

Docker has transformed modern software development, enabling the widespread reuse of containerized applications. Currently, Docker images are primarily distributed through centralized registries, among which Docker Hub is the largest, allowing developers to share and reuse images easily. The threats within these images also spread through the supply chain via dependency relationships, posing risks to anyone using the image and all images built based on it. However, it is unclear to what extent the threats within Docker images are distributed and propagated.

In this paper, we investigate five potential security risks in Docker images and propose a security analysis framework DITec­tor based on these security issues. We then utilize DITector to conduct a large-scale security measurement of the Docker image ecosystem. We collect descriptions of over 12 million image repositories from Docker Hub, construct an image dependency graph based on the layer information of the images, and select two sets of influential images based on their pull counts and dependency weight, totaling 33,952 images. Our findings are alarming: 93.7% of analyzed images contain known vulnerabilities, 4,437 images have secret leaks, 50 images contain misconfigurations and 24 malicious images. Furthermore, we identify 334 downstream images affected by malicious images and uncover patterns of attack propagation within the supply chain. We have discussed the measures to mitigate these issues, reported our findings to the relevant parties, and received positive responses.

## 1 Introduction

Docker [11] has revolutionized the way applications are developed, deployed, and managed, becoming an integral part of modern software engineering practices. It allows developers to package applications with their dependencies into portable containers for consistency across environments. This has resulted in complex supply chains where Docker images are frequently reused and built upon by various developers and organizations.

A Docker registry is a centralized system used to store, manage, and distribute Docker images, making it a crucial node in the Docker image ecosystem. Docker Hub [20] is the largest Docker registry, offering over 12.5 million official and community repositories, attracting more than 9 million users worldwide [6] to contribute over 20 billion image downloads per month [22].

In recent years, attacks on the Docker image ecosystem have become increasingly frequent. Attackers scan for leaked keys or software vulnerabilities in publicly available images, exploiting these flaws to carry out attacks [13, 31, 42]. Moreover, some attackers construct images that automatically execute malicious software, especially mining software, and use centralized Docker registries to spread these images, expanding the impact of the attack [1, 3, 4].

Several methods have been proposed to detect security threats in Docker images, primarily scanning for known vulnerabilities using public vulnerability datasets, including industry tools [2, 23, 43, 45] and academic work like DIVA [49]. Additionally, research has addressed other threats such as sensitive parameters, malicious files [9], and secret leakage [33]. However, existing methods are not consistent in terms of analysis dimensions, time, and scale. The selection patterns of datasets vary greatly, making it difficult to gain accurate and comprehensive insights into the security situation of the Docker image ecosystem through existing research. Moreover, most methods analyze each image individually, neglecting the inter-image dependency reuse relationships.

To address this issue, in this paper, we propose a large-scale security analysis framework called DITector (Docker Image Threat detECTOR), and systematically evaluate the security status of the Docker image ecosystem, covering five major security threats within images: software vulnerabilities, secret leaks, misconfigurations, malicious files, and sensitive parameters that might be included in the container startup commands provided by developers.

We collected information on over 12 million repositories, which, to our knowledge, constitutes the largest image dataset to date. Based on this, we built an image dependency graph, which allowed us to identify and analyze 33,952 critical images for the security of the Docker image ecosystem, including high-dependency-weight images that had not been discussed in previous work.

We found 4,437 images with leaked secrets, 50 with misconfigurations, and 24 malicious images. We investigated the propagation of various threats, revealed 334 downstream images affected by the malicious images, and examined intersections maintained by different users. We highlight the need for enhanced security measures and offer recommendations for the Docker community. We have reported the issues to stakeholders and received positive feedback. We will open-source part of the dataset and framework code to inspire future work on the security of the Docker image ecosystem.

In summary, we have made the following contributions:

- We propose a framework DITector to measure the security of the Docker image ecosystem. It identifies high-pull-count images and high-dependency-weight images and detects five types of security threats within them.
- We use DITector to conduct a large-scale measurement of the security of the Docker image ecosystem. Our measurement covers 33,952 critical images identified through information on over 12 million repositories.
- Our findings indicate that various threats are widespread within the Docker image ecosystem. We have disclosed our discoveries to stakeholders and call for measures to mitigate and prevent the emergence and spread of threats.

## 2 Background

### 2.1 Docker Image

Docker images are read-only templates for creating containers. An image is a software package that contains everything needed

to run an application. It consists of a series of read-only layers and the necessary configuration for the container. Developers can build and upload images to Docker registries, such as Docker Hub, to store and share customized application images. Users can use the `docker pull` command with an identifier formatted as *NAME[:TAG|@DIGEST]* to download the particular image.

Docker Registry manages images using a *repository-tag-image* hierarchy. Repositories contain multiple tags representing different image versions, and each tag may include images for various architectures and operating systems. An image is uniquely indexed by a digest, and the same image can be referenced by multiple tags. Docker registries are categorized into public and private, with Docker Hub being the largest public registry, containing over 12 million repositories.

In the process of developing, publishing, and sharing Docker images, three dimensions of information are introduced.

**Description.** Docker registries maintain descriptive information about images, incorporating both basic information introduced by the image itself and statistical information introduced by the Docker registry. Classified by hierarchy, the information can be divided into repository description, tag description, and image description. It involves important indicative details. For example, the repository description contains the repository's registration date, description, and pull count; while the image description includes instructions for creating each layer of the image.

**Content.** A Docker image comprises several ordered layers stacked on top of each other. Each layer represents a modification to the file system. All layers are eventually mounted by UnionFS to create a virtual file system visible within the container. The content includes the application, code, libraries, configurations, dependencies, system files, and other necessary components for running the application. However, introducing these contents also raises potential security concerns, such as vulnerable software, misconfigured applications, or accidentally packaged secrets.

**Metadata.** Image metadata consists of the basic information and configuration of an image. The configuration outlines the default settings and behavior for containers instantiated from this image, which contributes to the predictability and reproducibility of the containerized environment. However, this also introduces additional security risks. For example, the `ENTRYPOINT` and `CMD` are used to set the commands and arguments that the container executes, which could be exploited by attackers to launch malware.

### 2.2 Docker Image Supply Chain

The process of building images may introduce dependencies between images, forming the Docker image supply chain. It is a subset of the software supply chain, focusing on the dependencies between Docker images and all aspects of Docker images. In addition to the application, the image also contains all the content that supports the operation of the application. The newly built image is the child image (downstream image), and the dependent image is the parent image (upstream image). The child image entirely reuses the content and configuration of the parent image, resulting in the potential inheritance of risks from the parent image. Even if the parent image has patched the threat, due to delayed synchronization or content retention policies, the threat may still exist in child images built

before the repair and persist in Docker registries for an extended period. Therefore, having in-depth insights into the Docker image supply chain and ensuring the security of key image nodes in the supply chain is crucial to ensuring the security of the entire Docker image ecosystem.

## 3 Motivation and Problems

### 3.1 Motivating Case

Attackers are attempting to introduce complex call chains into images to bypass threat checks. We have identified a Docker image [51] more complex than typical malicious ones, introducing `docker-entrypoint.sh` and `baslat.py` on the basis of an image that installed XMRig [52]. It uses `ENTRYPOINT` to execute the Shell script, of which the filename is a common entry filename of many benign images. The script runs `baslat.py`, which in turn runs XM-Rig in a loop with a wallet address. The image also declares the opening of ports 443 and 17075 with `EXPOSE`, likely as a disguise.

We reported the findings to Docker Hub, which responded they would review and remove the malicious image. However, by the time we discovered the image, it had been uploaded for over two years with over 30,000 downloads. The current threat detection on Docker Hub is not comprehensive or timely enough. Therefore, in this paper, we conducted a security measurement on Docker Hub and assessed the security of the Docker image ecosystem.

### 3.2 Problem Scope

As the largest and public Docker registry, Docker Hub is a pivotal component of the Docker image ecosystem. It offers an extensive collection of official and community images for direct use or as bases for new ones. Therefore, our study focuses on Docker image security within Docker Hub. We assume attackers can build images and push the images to Docker Hub as normal developers. They can also pull, scan, and use public images as common users.

In this paper, we primarily focus on five types of threats present in the three dimensions of image information, including sensitive command parameters (in description), secret leakage (in description, content, and configuration), software vulnerability (in content), misconfiguration (in content), and malicious file (in content and configuration).

## 4 Methodology

To measure the security status of the Docker image ecosystem, we proposed DITECTOR, which utilizes a crawler to extensively fetch image description files from Docker Hub, extracts dependencies between images from the descriptions, and constructs IDEA (Image DEpendency grAph). DITECTOR identifies important images within the Docker image ecosystem based on image descriptions and IDEA. It downloads image content and metadata on demand for threat detection, aiming to discover security threats such as vulnerable software, misconfigurations, malicious files, and secret leaks within the images. The analysis framework is shown as Figure 1.

### 4.1 Data Collection

According to past research [12, 29], the average size of Docker images exceeds 100 MB (calculated to be 220 MB based on our data),

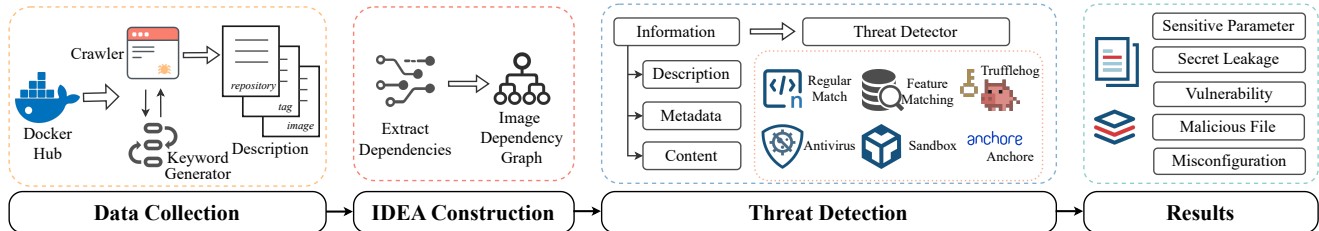

**Figure 1: Overview of DITECTOR.**

posing significant bandwidth and storage pressure for downloading and analyzing all images locally. Therefore, we first implement a crawler to collect the image descriptions. The descriptions include the pull counts of images in the repository and information about the image layers, which can guide us to identify important images in the ecosystem for further analysis.

The first step is to collect the names of a vast number of repositories. Docker Hub only provides an index file for official images [28]. For community images, Docker Hub offers an API [21] for retrieving repositories. The API accepts strings with a length between 2 and 255 characters as keywords and returns a maximum of 10,000 repository records for each keyword. We implement a depth-first search keyword generator to query community repositories. The generator generates the next search keyword according to the current keyword and the number of corresponding repositories. The crawler then obtains the repository, tag, and image descriptions with the name of each repository.

## 4.2 Image Dependency Graph Construction

The inheritance between images can be reflected in the image description, where the "layers" field of the child image contains that of the parent image. These layers consist of both content layers that include file modifications and configuration layers. The configuration layers only record the Dockerfile [24] commands executed to configure the image, and have no digests, with no file modification.

After collecting a substantial number of image descriptions, we can identify image dependencies. In this study, we proposed IDEA for constructing a Docker image dependency graph. IDEA uses Layer node to represent a layer within the image. Each Layer node can be uniquely indexed by the "id" attribute calculated through layer descriptions.

$$Layer_i.id = Hash(Layer_{i-1}.id + Hash(layer_{cont}.digest)) \quad (1)$$

$$Layer_i.id = Hash(Layer_{i-1}.id + Hash(layer_{conf}.command)) \quad (2)$$

The calculation methods for the id of different layer types are shown in Eq. 1 and 2. In the equation, "+" represents string concatenation, $layer_{cont}$ stands for content layer, and $layer_{conf}$ stands for configuration layer. For a content layer, IDEA uses its digest to calculate the Layer node id; for a configuration layer, it uses its instruction to calculate the id. The node id of the current layer is related to the ids of all layers below it so that the positional information of layers within the image is stored in the node id. We stipulate that $Layer_0.id$ is empty, which means that the id of the bottom layer of an image is calculated solely based on the current layer.

IDEA uses edge $Layer_i-> Layer_j$ to indicate that $Layer_i$ is the direct underlying node of $Layer_j$ in a certain image. It marks the image name on the Layer node corresponding to its top layer. Therefore, we can search upstream or downstream images of a specified image along the chain of Layer.

## 4.3 Threat Detection Method

Based on the image descriptions and IDEA, we can identify image nodes that are critical to the image ecosystem. We have identified two types of critical nodes: (1) High-pull-count images. The security threats within will directly affect the users of the images; (2) High-dependency-weight images, where the security threats within may propagate downstream to a wide range of images.

A threat detector is implemented in DITECTOR to analyze these images. The detector reuses existing detection results, including analyzing only the descriptions that differ from previously analyzed images with the same digest; for image content, the detector scans for threats from the bottom layer to the top layer, reusing the results of already scanned layers. The detector identifies five types of security threats in different dimensions of the image information.

**Sensitive Command Parameter.** It uses regular expression to extract the startup commands provided by developers from the repository description and identifies five categories of sensitive parameters that we have manually reviewed from the manual [19] (see Appendix B), which cover major container security restriction that might be bypassed through parameters: (1) breaking filesystem isolation; (2) breaking network isolation; (3) breaking process isolation; (4) breaking resource limitations; and (5) elevating container runtime privileges.

**Secret Leakage.** Similar to prior work [9], we utilize Truffle-Hog [47] to scan the leaked secrets within the images. The detector uses TruffleHog to directly scan descriptions and configuration files, and scan the content layer by layer.

**Software Vulnerability.** The detector uses Anchore [2], which has been employed in previous work [18, 35], to perform Software Composition Analysis (SCA) and vulnerability detection on images. Anchore utilizes datasets like the NVD [38] as vulnerability sources and supports the detection of all layers within the image.

**Misconfiguration.** We manually reviewed the configuration manuals and identified insecure configurations for four popular containerized database applications (including MongoDB, Redis, CouchDB, and Elasticsearch), focusing mainly on unauthorized access. The detector identifies configuration files within layers based on keywords in the path and detects misconfigurations.

**Malicious File.** For files executed by containers, the detector locates the file path according to the image configuration and uses an antivirus engine provided by our corporate partner, which is a large cyber security company. Regarding malicious components installed from third-party registries, we first established datasets for malicious PyPI and npm packages, including those identified by our corporate partner, and datasets established by Guo et al. [17] and OpenSSF community [40]. The detector extracts components installed by package management tools like pip and npm from the image description and matches it against the dataset of malicious packages. For suspicious components, it verifies maliciousness based on file characteristics.

### 4.4 Implementation

We implemented the web crawler using Python's Scrapy [44] package. We leveraged the Goroutine and Channel features of Golang to implement the multi-threaded IDEA builder and the threat detector. We used MongoDB [36] to store the collected descriptions and detection results. We employed Neo4j [37] to store IDEA and execute graph algorithms to calculate the image dependency weights and identify upstream and downstream images of a specific image.

## 5 Evaluation

With the support of a wealth of image descriptions and IDEA, we can evaluate the security status of the Docker image ecosystem from a more macroscopic perspective. In this section, we outline the experimental design, present the experimental results, and analyze the findings to answer the following questions.

**RQ1:** How are images distributed in the image ecosystem?

**RQ2:** How is the distribution of various threats in the Docker image ecosystem?

**RQ3:** How do threats propagate downstream along dependency relationships, and to what extent?

### 5.1 Dataset

We adopted the crawler for a total of six months, starting in August 2023, until the number of repositories stabilized and no longer increased significantly. Finally, we collected a total of 12,079,309 repository descriptions and 29,775,651 tag descriptions, including 175 official repositories and 117,676 tags associated with them, which is the largest image dataset to our knowledge.

For the sake of efficiency in building the dependency graph, we pre-selected two types of images with the most basic influence for the Docker image ecosystem security to construct IDEA: (1) all official images and (2) the top 10 most recently updated images from repositories with at least 100 pulls. The final dependency graph includes 433,613 official images and 3,238,990 community images, consisting of 24,739,338 `Layer` nodes.

### 5.2 Docker Image Distribution (to RQ1)

We calculated the distribution of images based on the pull count of repositories, as shown in Figure 2. The points on the line represent the minimum, quartile, median, third quartile, and maximum values. The results indicate that official repositories have much higher pull counts than those of community repositories. However, since official images only account for less than 0.002% of all images, the distribution of all repositories is almost identical to that of community repositories. The most pulled repository is `library/alpine`, with over 10 billion downloads. There are more than 90% repositories with pull counts of less than 100, indicating that they are only lightly used by relevant organizations after being uploaded. Among them, 4,405,608 repositories were never pulled, which means that images within over 36% of repositories have never been used even once after being uploaded.

We explored along the edge described in Section 4.2 on IDEA to calculate the dependent weight (the number of images it depends on) and dependency weight (the number of images that depend on it) of each image in IDEA. Besides, since the same image can be tagged multiple times, we deduplicated images that have upstream or downstream images based on their digests and also deduplicated upstream images based on their digests when calculating the dependent weights. We did not deduplicate downstream images, as threats may propagate to images with the same digest in different repositories. The results are shown in Figure 3 and 4. In these figures, ID represents the In-Degree (dependent weight), and OD represents the Out-Degree (dependency weight).

The results show that over 80% of considered images have at least one upstream image. Meanwhile, over 90% of the images do not have downstream images. More than 75% of images with downstream images have less than 10 downstream images. These results demonstrate that the majority of images rely on a very small proportion of all images. The images with high dependency weights are crucial for ecosystem security. However, due to the limited data scale, previous work [7, 9, 33, 49, 55] did not analyze these images.

Therefore, we identified two categories of images with significant impact for in-depth analysis: (1) **High-pull-count images.** The top 3 most recently updated tags for repositories with pull counts greater than or equal to 1 million, which are likely to be heavily used directly. (2) **High-dependency-weight images.** The images with dependency weights greater than or equal to 10, where threats within these images may propagate to a larger number of downstream images. For high-dependency-weight images, we deduplicated the images based on the image digests. Due to Docker's lack of support for downloading images with only digest, we selected a representative image name for download detection among all names. This means prioritizing official repositories or the earliest updated tags, which are more likely to still be hosted in the registry. Finally, we identified 20,673 high-pull-count images and 25,924 high-dependency-weight images.

### 5.3 Threat Distribution (to RQ2)

DITECTOR downloaded high-pull-count images and high-dependency-weight images via names with digests, and then analyzed these images using the threat detector. We found that there are 151 intersections between the two sets of images, and during our process of crawling descriptions and analyzing images, 12,494 images became unavailable, due to reasons including images being deleted, developers rebuilding new images to replace the original ones, etc. 75% of the expired images belong to high-pull-count image set. In the end, we analyzed 33,952 images. In this section, we evaluate the detection results of various threats.

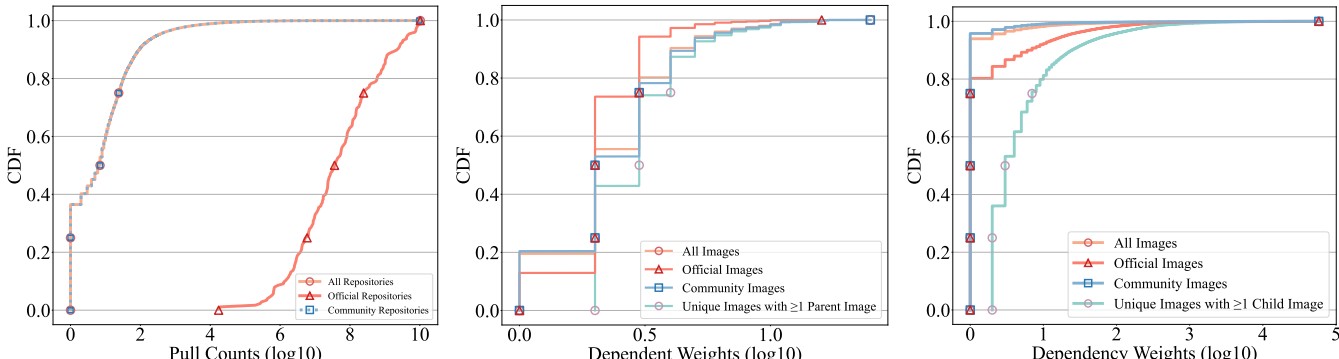

Figure 2: CDF of repository pull count.   Figure 3: CDF of image ID.   Figure 4: CDF of image OD.

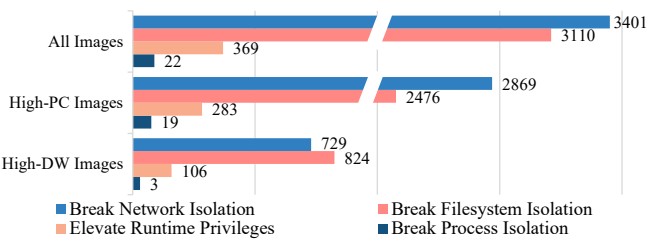

Figure 5: Distribution of each sensitive parameter.

*5.3.1 Sensitive Command Parameter Distribution.* We identified sensitive parameters in the startup commands in repository descriptions. The distribution of sensitive parameters of classified statistics is shown in Figure 5, where High-DW represents high-dependency-weight and High-PC represents high-pull-count.

The detected images come from 10,741 different repositories. Among them, 1,795 images provided a startup command with parameters in their descriptions, accounting for 16.7%. Of these, 78.4% of the images included sensitive parameters in the command. This suggests that most developers do not provide usage commands for their images, while repositories that do provide startup commands are likely to use sensitive parameters in their commands.

It is worth noting that although we identified five sensitive parameters, we did not find parameters that are used to break resource isolation, represented by `-cgroupns`, which supports running containers in the cgroup namespace of the host. This indicates that developers expect containers to run in resource-constrained environments, and few users would take on the risk of a denial-of-service attack introduced by parameters.

Although studies warn of serious security issues from using `-privileged` to provide capabilities for containers [10, 41], we found 103 repositories using it. Further investigation revealed that these repositories are mainly used in scenarios where additional privileges do need to be applied for containers, such as monitoring the host, building images across platforms, and running Docker in Docker. However, the `-privileged` may grant excessive permissions, increasing the risk of container escape. It is recommended to use `-cap-add` to explicitly add specific kernel capabilities as needed [25].

*5.3.2 Secret Leakage Distribution.* We discovered leaked secrets in 29,420 images, accounting for 86.7% of the images analyzed. Since research has shown that TruffleHog has a low precision [5], to reduce false positives caused by ambiguous rules and example secrets, we verified the leaked secrets under the ethical constraints described in Appendix A without actually calling API.

**Precise Regular Expression (filter Imprecise).** Upon reviewing the source code, we filtered secret detectors from TruffleHog that contain at least one exact rule. It means that the selected regular expressions are not generated by the built-in heuristic rule generation method `detectors.PrefixRegex` and exhibit distinct features. We ultimately selected 46 precise regular expressions related to 38 applications, including private keys, URIs (which contain usernames and passwords), and API tokens. We further used the expressions to filter the results.

**Common Pattern (filter Pattern).** Different types of secrets have different common patterns that serve as templates or examples. For private keys, TruffleHog ensures that the detection results are parseable. Following the method outlined by Dahlmanns et al. [9], we excluded private keys recorded in the kompromat project [48] that had already been deemed public. For URIs, we did not find a public dataset of example username-password pairs. Thus, we used ChatGPT [39] to generate example username-password pairs and excluded URIs with default addresses and common example pairs. For API tokens, previous research has shown that example tokens may be manually crafted [34], consisting of human-readable characters such as `"EXAMPLE"`, or patterned characters like `"XXXXX"` or `"ABCDE"`. Therefore, we excluded tokens that contain words with example meanings, or substrings with a length of five or more identical characters or sequential characters.

**Leakage Frequency (filter Frequency).** Two primary scenarios can lead to a valid secret being detected multiple times across various images: (1) The secret is leaked in a layer of an upstream image, causing the secret to be also detectable in its downstream images. (2) The secret is packaged multiple times by the same developer and released in different images. We consider secrets in other cases to be either example keys or invalid keys. Therefore, for each type of secret, we calculate the average number of times each secret is detected (leak count) and the average number of layers that contain the secret (leak layers). We consider a secret to be invalid if the images that contain it come from at least 2 different users and meet

either of the following conditions: $leak\ count \geq 2 * leak\ count_{avg}$, or $leak\ layers > leak\ layers_{avg}$. We excluded such invalid secrets with a relatively high leakage frequency.

**Table 1: Exclusion and distribution of leaked secrets.**

| Secret Type | Invalid | | | Valid |
|---|---|---|---|---|
| | Imprecise | Pattern | Frequency | |
| **Private Key** | 0 | 71,256 | 35,752 | 10,570 |
| **URI** | 1,618 | 716,314 | 272,468 | 24,888 |
| **API Token** | 4,718,131 | 24,853 | 48,027 | 7,494 |

The exclusion and distribution of the secrets are as shown in Table 1. Eventually, 99.3% of the detected secrets are invalid. We verified 42,973 secrets which are distributed in 4,437 images. 24.8% of the high-pull-count images and 7.3% of the high-dependency-weight images contain leaked secrets, indicating that high-pull-count images are more prone to leaking secrets compared to high-dependency-weight images. The distribution of three types of secrets is similar in both types of images. URIs account for more than half of the leaks, followed by private keys, with API tokens being the least common.

Valid secrets have been found in the description, metadata, and content of images. Among them, more than 99.6% of the secrets are leaked through the image content. We found 122 secrets in the image description, which indicates that attackers can obtain valid secrets at a relatively low cost only by crawling and scanning a large amount of image descriptions through the Docker registry API, without actually downloading the images. These secrets are all leaked in the instructions within the layers field of the image description. All the secrets we found in the image metadata (a total of 21) can be detected in the image description because they are all set as environment variables through the ENV instruction. This means that some developers tend to write secrets into environment variables as they would in a host environment, which is insecure for images and increases the risk of exploitation by attackers. We have disclosed the findings to the image maintainers, with details visible in Section 6.3.

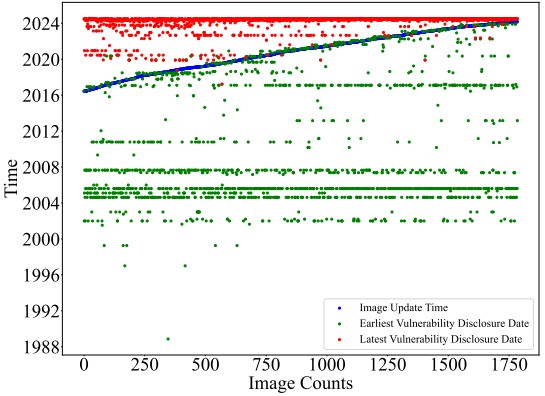

**Figure 6: Image update time and vulnerability published time within.**

### 5.3.3 Software Vulnerability Distribution.
Considering the efficiency of the detection, we randomly selected 1,000 high-pull-count images and 2,000 high-dependency-weight images from the datasets based on their quantity ratio for vulnerability detection. The results show that all high-pull-count images contain software vulnerabilities, with an average of 558 vulnerabilities per image, including more than 17 critical vulnerabilities. In contrast, 90.6% of high-dependency-weight images contain software vulnerabilities, but each image contains an average of 1,022 vulnerabilities, including more than 29 critical vulnerabilities. The average severity of vulnerabilities in high-dependency-weight images is much higher than that in high-pull-count images.

To investigate the impact of image update times on vulnerability distribution, we further scanned the most recently updated images in the repositories of high-dependency-weight images. In the entirety of the 796 scanned images (with different images coming from the same repository), 158 images belong to the high-dependency-weight image set. The corresponding repositories have not been updated after 2020, with an average of 788 known vulnerabilities per image. This indicates that over a decade of development, many foundational images on Docker Hub are either poorly maintained or have been deprecated. The potential issues arising from the vulnerabilities they contain deserve attention.

Studies have indicated that the number of vulnerabilities contained in images is related to the operating systems they are based on [35]. Therefore, we arranged the images of the same repository according to the upload time and calculated the repair of vulnerabilities based on the operating system. The results indicate that the new images released in image repositories have a significant repair situation for the vulnerabilities of the old images. On average, each new image fixes 164 vulnerabilities, which is 35% more than the number of vulnerabilities introduced. 60% of the new images fix more vulnerabilities than they introduce, and 41% of them fix existing vulnerabilities without introducing new ones.

For Docker images with complete time records, we compared the update time of the images with the earliest and latest published times of vulnerabilities within the images. The results are shown in Figure 6. Among them, 83.5% of the images contain at least one vulnerability that was published in 2024, which reminds developers that they should regularly update the images to reduce the impact of vulnerabilities. We found that 11.1% of the images were updated before the disclosure time of the earliest vulnerability within them. This means that these images did not contain any known vulnerabilities at the time of their release. Additionally, we found that vulnerabilities from 30 years ago may still exist within images. An image updated in 2018 contained a critical vulnerability that was published in 1988, and the repository for that image has ceased maintenance since that update. This alerts users to thoroughly inspect images before using them, ensuring they understand and can mitigate any security risks contained within.

### 5.3.4 Misconfiguration Distribution.
We identified 557 misconfigurations in 398 images. Since the configuration files packaged into the image may not take effect in the container, we first checked whether the files executed at container startup would run the corresponding applications. The results show that only 50 images run the application, containing 103 files with misconfigurations. We

audited the configuration files and found that over 93% of the misconfigurations were unauthorized access issues caused by reusing the official example configuration files of the related applications.

To assess the vulnerabilities of misconfigured images in real-world scenarios, we identified services that might be created and exposed using specific images based on data from our partner's cyberspace mapping platform. Since users can publish a container port to any host port, we filtered and validated the results based on the distinct characteristics in the configuration files, without actually exploiting the target systems. For instance, we found that the image `dtagdevsec/elasticsearch:dev` sets `xpack.security.enabled` as false in the default configuration file of ElasticSearch, which will disable the security features, leading to unauthorized access. It also declares the cluster name as `tpotcluster` and the node name as `tpotcluster-node-01`. We discovered that the response from port 9200 of an IP address from Germany matches the characteristics exactly, indicating that the service is likely implemented based on this image. Further investigation revealed that the image points to a honeypot project called T-Pot [46], where users are aware of and accept this vulnerability exposure. We also found ports open for services such as ElasticSearch and Redis that match the unique configurations within other images. These findings suggest that images with misconfigurations may have been used in the wild. Users may face unauthorized access issues and potential data breaches if they deploy containers using these images in production environments without checking their specific configurations.

*5.3.5 Malicious File Distribution.* The default executed files from 31 images were detected as malicious, including 26 cryptocurrency mining software or scripts, 4 proxy agents, and 1 adware. Among the mining images, 15 directly execute XMRig, an open-source mining software that can mine multiple cryptocurrencies, 6 use scripts to set up XMRig, and 5 execute other mining software.

We inspected whether containers launched based on the image would exhibit malicious behavior. For images that execute mining software or scripts, we reviewed the image metadata, script files, and configuration files of the software. We consider the image configuring the mining pool or wallet for the mining software as malicious, as the cryptocurrency mined by the containers will profit the image maintainer. For other malicious software, we extracted them from the image and ran them in a dedicated sandbox to verify whether they showed malicious behavior.

We verified that 24 images are malicious, all of which are used for mining. Among them, 18 configure the software by setting CMD, 5 configure it in the executed script, and 1 configures it through a configuration file. Some findings are noteworthy. We found that user `servethehome` removed the configuration for mining software in the latest image of the repository `monero_cpu_minergate`, repurposing it as a template image only used as a mining tool. Besides, we found that the same wallet address is used in the images maintained by different users, such as `ngaymaisang/ngaymaisang:latest` and `thanhcongnhe/thanhcongnhe:latest`. We further analyzed other images maintained by the users and found that among the 11 images maintained by namespace `thanhcongnhe`, 9 execute a malicious Python script named `dao.py`. The total pull count of these images has been close to 100 million. Four kinds of mining software are used in these images to mine three kinds of cryptocurrencies:

Monero (XMR), PacketCrypt (PKT), and Crypton (CRP). A specific example can be found in Appendix C.

We tracked the PKT wallet and found that it started to generate transactions right after the image was uploaded. The final income was 594,615 PKT, approximately worth $900. We were unable to track the XMR and CRP wallets due to their privacy policies. However, we observed a study [8] that revealed a batch of malicious images, which have been removed from Docker Hub, using the same wallet addresses as the malicious images we discovered. As of 2020 when that study was released, the wallet had already mined 525.38 XMR, which is now worth more than $90,000. This indicates that the attacker has long begun to upload malicious images to abuse the resources of the victimized host for mining and profiting. Meanwhile, they registered different user names and repository names to increase the scope of the attack and bypass supervision.

Additionally, no malicious packages installed from PyPI or npm were found in the considered images. This indicates that attackers are more inclined to directly implant custom malicious code into Docker images rather than introduce other third-party repositories to store malicious code.

## 5.4 Threat Propagation (to RQ3)

Based on the results of threat detection, we investigated the propagation of various threats by searching downstream images of the images containing threats on IDEA. Considering that descriptions of upstream images are not reused, we only focus on the threats that can be exposed in the image metadata and content.

Among images containing leaked secrets, 1,879 images have at least one downstream image, of which over 93% are inherited by images from different repositories. Due to the inheritance mechanism of images, secrets cannot be fixed by new layers of the image. Furthermore, the image may contain context that uses the secrets, allowing attackers to easily exploit the secrets to launch attacks. This means that the secrets have already spread outwards and solidified into images maintained by other users, increasing the possibility of the secrets being exposed.

The threats, including software vulnerabilities, misconfigurations, and malicious files, can be mitigated at the top layer by fixing the issues present in the lower layers. A significant finding is that the maintainers of the downstream images built from images containing malicious files are well aware of the mining software installed in the upstream images. They have made mining configurations that benefit themselves during inheritance and have created a large number of repositories to expand their influence.

We have found 2 malicious images have downstream images. One of them has 333 downstream images. By examining the digests of images in different namespaces, we identified four groups of Docker Hub users, within each of which at least one identical image was maintained by different users, implying that the users in each group might be controlled by a single maintainer. For each set, we manually analyzed the same images uploaded by different users and confirmed that all intersecting images were configured with mining pool addresses or wallet addresses.

Based on the distribution of images within different groups, it can be observed that different attackers have different choices between creating more users or more repositories. However, overall, only

a small number of images, usually just one, are uploaded to each repository. For example, in one group we found that there were 192 images across 4 namespaces, with a single user named `rini002` creating 127 repositories. Other attackers prefer to create a large number of different users. In a group containing 11 namespace groups, there were only 18 images, with 8 namespaces having created only one repository each. The total downloads of these images had exceeded 50 million, which may have already had a huge impact. We reported our findings to the Docker Security Team, as can be seen in Section 6.3.

## 6 Discussion

### 6.1 Limitation

As the largest, public and default registry, Docker Hub contains complex interdependencies that make it a representative sample of the Docker image ecosystem. Therefore, we focused our analysis on Docker Hub images. A substantial amount of prior work has systematically analyzed the distribution of vulnerabilities within images, and, additionally, most software vulnerabilities reported by current tools are often not exposed or exploited within containers. Therefore, we selected a subset of images from the dataset proportionally, scanned their software vulnerabilities, and analyzed recently updated images in the repositories of the high-dependency-weight images to evaluate the software vulnerability fix cycle.

Moreover, we used several open-source tools, such as TruffleHog and Anchore, to detect threats in the images. The accuracy, completeness, and reliability of our results are limited by the functionalities and capabilities of these tools. Future work could benefit from the development of more advanced, customized tools to enhance detection accuracy and reduce reliance on third-party solutions.

### 6.2 Mitigation

For image users, it is essential to use security tools to thoroughly check for software vulnerabilities, malicious files, and other security issues in the image before use. When deploying services such as databases in containers, it is important to check the configuration files of the image to prevent security issues like data leaks caused by unauthorized access configurations.

Image maintainers should regularly update images and promptly fix vulnerabilities to reduce the risk of exposure. Besides, it is important to use secret scanning tools to check whether the image contains any valid secrets before publishing. This helps prevent secrets from being leaked into the supply chain with the image, where they could be discovered and exploited by attackers.

Docker registries should provide more security information for users. Currently, popular image registries such as Docker Hub only support SCA and vulnerability scanning for images. However, there are still many other security risks in Docker images, such as privacy leaks and malicious files, which can pose significant security threats to both image maintainers and users. Comprehensive security analysis of images by Docker repositories is crucial for ensuring the security of user systems and the security of the image ecosystem.

### 6.3 Responsible Disclosure

We disclosed our findings to stakeholders. For images leaking secrets, considering that the secrets exposed for a long time have highly probably become obsolete, we extracted email addresses and relevant GitHub project URLs from the repository descriptions and layer instructions of the images created within one year before the end of our crawling period, totaling 134, and attempted to contact them. We received over 50 responses, confirming their status as maintainers, expressing gratitude for our work, acknowledging and fixing the secret leakage issues, or informing us that the secrets were part of the container's operation or were no longer in use.

For malicious images, we reported the list of discovered images and the attack mechanism to the Docker Security Team. We received a response appreciating our efforts, informing us that they would analyze those images and whether this constitutes a large-scale attack, and that they would take down the malicious images.

## 7 Related Work

Significant research has been conducted on Docker security. One direction is to study the containers runtime security [27, 50, 53, 54], including privilege escalation [30] and bypassing resource limits [14–16]. Some studies focus on image security in Docker registries. Shu et al. [49] proposed DIVA employing Clair to detect vulnerabilities in images from Docker Hub. Zerouali et al. [55] assessed the vulnerabilities and bugs in Debian-based images. Liu et al. [33] analyzed sensitive parameters, malicious files, and vulnerabilities in images. Liu et al. [32] investigated typosquatting attacks on Docker registries. Dahlmanns et al. [9] studied the leakage of private keys and API tokens in Docker registries and analyzed the use of leaked secrets in the wild.

However, the analysis dimensions, time, and scale of the datasets of these works are inconsistent, making it difficult to form a comprehensive understanding of the security status of the entire Docker image ecosystem. Besides, previous work did not consider the dependency relationships of images in the selection of the image dataset and did not analyze the high-dependency-weight images in the ecosystem. These have prompted our work to complement previous efforts, providing a more comprehensive view of the security of the Docker image ecosystem.

## 8 Conclusion

In this paper, we propose a large-scale Docker image security analysis framework, DITECTOR, to systematically investigate the security of the Docker image ecosystem. We collected extensive data from Docker Hub, built a dependency graph IDEA, and identified two key image categories: high-pull-count images and high-dependency-weight images. We analyzed five types of threats within the images and assessed the spread of threats based on IDEA. Our findings show that threats are widespread and can propagate downstream along the supply chain. We shared these insights with stakeholders, receiving positive feedback. This study highlights the need for improved Docker ecosystem security and provides recommendations for image maintainers and the community.

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

## A  Ethical Considerations

We take ethical considerations seriously and adopt ethical guidelines [26, 56] in our work. First, we restrict the access rate of the crawling and downloading process to comply with Docker Hub's rate limits, ensuring it does not disrupt the normal operation of Docker Hub. Second, we do not create vulnerabilities. All the information we use is publicly available, and all vulnerabilities discovered were exposed before being detected. Moreover, we refrain from hosting vulnerable images in public repositories. Instead, we upload images only to private repositories under our own account and remove them after experimentation. Third, we conduct the validation process locally in a controlled environment. This means we do not verify leaked secrets by actually calling with them, which could potentially impact victims. We also do not create online containers to validate vulnerabilities, which might increase the risk of attacks on the cloud platforms. Fourth, upon confirming the threats, we have promptly disclosed our findings to the corresponding parties and received feedback. Furthermore, we take measures to ensure all crawled data and analysis results are stored in controlled environments and are not leaked. These measures include running experiments on hosts that are accessible only through MFA (Multi Factor Authentication) login and not exposed to public networks, and ensuring databases can only be accessed through strong passwords.

## B  Sensitive Parameters

Docker allows users to break resource isolation or capability restrictions by passing parameters when starting containers, enabling containers with specific needs to work correctly. However, containers launched with these parameters can also provide convenience for attackers. We manually reviewed the `docker run` manual [19] and identified five categories of sensitive parameters: (1) breaking filesystem isolation; (2) breaking network isolation; (3) breaking process isolation; (4) breaking resource limitations; and (5) elevating container runtime privileges. Details are as shown in Table 2.

**Table 2: Identified sensitive parameters.**

| Parameter | Description | Category |
|---|---|---|
| –mount | Mount a filesystem | 1 |
| –volume | Bind mount a volume | 1 |
| –network | Connect to a network | 2 |
| –publish | Publish a container's ports | 2 |
| –publish-all | Publish all exposed ports | 2 |
| –ipc | IPC mode | 3 |
| –pid | PID namespace | 3 |
| –cgroupns | Cgroup namespace | 4 |
| –device | Add a host device | 5 |
| –cap-add | Add Linux capabilities | 5 |
| –privileged | Add extended privileges | 5 |

## C  Malicious Images

We discovered that 9 images maintained within the namespace thanhcongnhe were engaged in cryptocurrency mining using a Python script named dao.py. For instance, the image thanhcongnhe /lancuoicung:latest, which has been pulled over 4 million times, downloads code files from the remote server and saves them in the image. The Dockerfile command is shown in Listing 1, and the code snippets are shown in Listing 2. The code first starts XMRig in the background to mine XMR, and then starts PacketCrypt to mine PKT.

**Listing 1: Command for introducing malicious code.**

```
1 /bin/sh -c wget https://raw.githubusercontent.
      com/giautoidi/giautoidi/beta/dao_cpu_100.py
      -O /etc/dao.py
```

**Listing 2: Code snippet for running mining software.**

```
1  # Download and update XMRig...
2  command_xmrig_default = '--algo randomx -o
      45.8.146.102:443 -u 43
      ZBkWEBNvSYQDsEMMCktSFHrQZTDwwyZfPp43
      FQknuy4UD3qhozWMtM4kKRyrr2Nk66JEiTyp
      fvPbkFd5fGXbA1LxwhFZf -p nql --tls --cpu-
      max-threads-hint=100 --http-host=0.0.0.0 --
      http-port=80'
3  command = '/opt/%s/%s %s' %(folder_xmrig,
      xmrig_name, command_xmrig_default)
4  if os.path.isfile('/usr/bin/screen'):
5      os.system ('screen -dmS %s %s' %(xmrig_name
          , command))
6  elif os.path.isfile('/usr/bin/nohup'):
7      os.system ('nohup %s &' %command)
8  else:
9      os.system ('%s &' %command)
10
11 # Download and update PacketCrypt...
12 command = '/opt/%s ann -p
      pkt1qhwf4s4d8dvzev9dc4l7qxz8v0tpetfw6s 5
      h0uv http://pool.pkteer.com http://pool.
      pktpool.io/ http://pool.pkt.world/' %
      pkt_name
13 if os.path.isfile('/usr/bin/screen'):
14     os.system ('screen -dmS %s %s' %(pkt_name,
          command))
15 elif os.path.isfile('/usr/bin/nohup'):
16     os.system ('nohup %s &' %command)
17 else:
18     os.system ('%s &' %command)
```

