# OpenReview forum: "Dr. Docker: A Large-Scale Security Measurement of Docker Image Ecosystem"
_ACM.org/TheWebConf/2025/Conference — WWW 2025 Poster_

### Official Review · Reviewer_BHdC · 2024-11-03

**Novelty:** 3
**Technical Quality:** 2

**Review:**

# Summary

This paper presents a comprehensive security measurement study of the Docker image ecosystem through a proposed framework called DITector. The authors analyze over 12 million Docker Hub repositories to investigate five types of security threats: software vulnerabilities, secret leaks, misconfigurations, malicious files, and sensitive parameters. They construct an image dependency graph (IDEA) to understand threat propagation patterns and identify critical images based on pull counts and dependency weights. The evaluation covers 33,952 critical images and reveals alarming security issues including widespread vulnerabilities (93.7%), numerous secret leaks (4,437 images), misconfigurations (50 images), and malicious images (24) primarily used for cryptocurrency mining. The study also tracks threat propagation, identifying 334 downstream images affected by malicious images.

# Strengths

* Analyzes over 12 million repositories, representing the largest dataset in this field.
* Covers multiple dimensions: image descriptions, content, and metadata
* Introduces DITector, a security analysis framework
* Identifies concrete security issues affecting real-world systems
* Promises to open-source part of the dataset and framework code

# Weaknesses
* No clear rationale for focusing on these specific five types of threats.
* DITector primarily integrates existing tools (TruffleHog, Anchore, antivirus engines) without significant technical innovation. Hence the main contribution appears to be integration rather than novel technical solutions. Hence the main contribution appears to be integration rather than novel technical solutions.
* Results primarily focus on statistical findings without deeper insights.
* Recommendations are too general and missing analysis of interesting patterns or unexpected discoveries.
* Misconfiguration only includes  four popular containerized database applications and completely overlooks misconfigurations in other types of containerized applications. Different applications have distinct security requirements and potential misconfiguration patterns. By focusing solely on database applications, the study fails to capture the diverse range of configuration vulnerabilities in the Docker ecosystem.This narrow scope undermines the credibility of the evaluation results.

Despite the large-scale analysis, the paper's findings are neither surprising nor insightful, and its recommendations remain generic rather than offering specific, actionable solutions to Docker ecosystem security. Moreover, the study's scope is concerningly narrow, focusing on only five types of security threats and examining misconfigurations in just four database applications. This limited coverage makes it difficult to accept the generalizability of the experimental results across the broader Docker ecosystem. For a paper claiming to conduct a comprehensive security measurement study, these constraints significantly undermine its contribution and conclusions. I therefore cannot recommend it for publication at The Web Conference.

**Questions:**

* Why you choose five types of threats listed in Section 3.2, why other potential security threats were excluded.
* What is the connection between this paper and Web ?
* Why only only includes  four popular containerized database applications for Misconfiguration ? Why other misconfiguration are filtered.
* What is the specific distribution of the 'over 50 responses' across different response categories, and will these responses be included in the open-source version of the dataset?

**Reviewer Confidence:**

3: The reviewer is confident but not certain that the evaluation is correct

**Scope:**

3: The work is somewhat relevant to the Web and to the track, and is of narrow interest to a sub-community

---

### Official Review · Reviewer_Jv8d · 2024-11-22

**Novelty:** 4
**Technical Quality:** 6

**Review:**

I would like to thank the authors for their work. Dr. Docker is a framework for security analysis of the Docker image ecosystem. The authors launched a large-scale security measurement, looking for images with known vulnerabilities, containing leaked secrets or other types of risks. One distinguishing aspect of this paper compared with prior work is that it pays attention to inter-image dependency reuse relationships.  The authors also propose mitigations for some of the risks that were found.

The methodology used seems sound, and the authors found a considerable number of security issues, overall the paper seems relevant and impactful.

Unfortunately, I do not see any significant connection between this paper and the web, making me doubt how much this paper fits in the web conference.

Writing quality throughout the paper is reasonable, but every so often there is a sentence with confusing language:
-	I would move the explanation of XMRig in Section 5.3.5 back to the first place where you introduce it.
-	Weirdly phrased sentence: "There are more than 90% repositories with pull counts of less than 100"
-	"We adopted the crawler for a total of six months"
-	"However, we observed a study [8] that revealed (...)"

**Questions:**

1. You committed to open source part of your tool and dataset, but it is unclear to me what exactly are these parts and what is not going to be open sourced.

2. Can the authors provide more details with respect to how the "keyword generator" algorithm is implemented?

3. "We received over 50 responses, confirming their status as maintainers, expressing gratitude for our work, acknowledging and fixing the secret leakage issues, or informing us that the secrets were part of the container’s operation or were no longer in use."
-    Can you describe the frequency of each of these responses, out of the 50? Including this breakdown in the paper could be of value.

**Reviewer Confidence:**

3: The reviewer is confident but not certain that the evaluation is correct

**Scope:**

2: The connection to the Web is incidental, e.g., use of Web data or API

---

### Official Review · Reviewer_3ejN · 2024-11-27

**Novelty:** 4
**Technical Quality:** 3

**Review:**

This paper proposes a new Docker image security detection framework, **DITector**, aimed at analyzing images and their security risks in Docker Hub. By collecting image description files, the authors construct a Docker Image Dependency Graph (IDEA) and use this graph for comprehensive security analysis of the images. The framework can detect five types of security threats: sensitive command parameters, secret leakage, software vulnerabilities, configuration errors, and malicious files. The authors also describe how existing tools (such as TruffleHog and Anchore) are utilized to detect these threats and improve detection efficiency through iterative analysis.

Strengths:

1. **Innovation in Technical Framework and Methodology**:
   - The IDEA (Docker Image Dependency Graph) presented in this paper uses the structure of graphs to represent the dependencies between images, offering a new perspective for image security detection. Notably, the hierarchical relationships establish an inheritance chain from parent images to child images, effectively reflecting the hierarchical dependencies and security propagation mechanisms of images.
   - For dependency analysis, the authors use graph computation methods to calculate layer IDs and identify key images within the ecosystem, focusing security checks on these images. This dependency-based analysis method is the first of its kind in the existing literature and is highly innovative from a technical standpoint.
   - The design of DITector also demonstrates an efficient detection framework by reusing previous detection results (such as images with the same digest), optimizing performance. This incremental threat detection method enhances the framework's practical usability.

2. **Experiments and Results**:
   - The experimental section comprehensively validates DITector, showing that the framework effectively identifies various security threats present in Docker Hub images. The authors also provided feedback on insecure images, receiving corresponding responses, which demonstrates practical application value.

Weaknesses:

1. **Lack of Comparative Analysis**:
   - While the paper mentions using existing tools for threat detection, the comparative analysis is not detailed enough. It is recommended to further compare DITector with other Docker image security detection tools in terms of accuracy, performance, and other metrics, highlighting the innovations and advantages of this approach.

2. **Future Directions for Improvement**:
   - The paper does not discuss defense strategies for the existing security threats in Docker images. This includes the role of current defense methods in mitigating security risks in Docker images and potential novel strategies for addressing these threats. A discussion on defense mechanisms could enhance the paper’s completeness.

**Questions:**

1. **Comparison with Existing Tools**:
   - How does DITector compare to other existing Docker image security detection tools in terms of accuracy, performance, and scalability?
   - Can the paper include a more detailed comparison between DITector and these existing tools to highlight the innovative aspects and advantages of DITector?

2. **Lack of Defense Strategies**:
   - What defense strategies could be applied to mitigate the security risks identified by DITector and how about their performance?
   - Could the paper discuss novel defense mechanisms or strategies to address the identified security threats?

3. **In-depth Experimentation and Results Analysis**:
   - How does DITector’s performance vary with large-scale datasets or more complex images?

4. **Generalization to Other Container Platforms**:
   - Can the DITector framework be extended to analyze other container platforms beyond Docker, and if so, what adjustments would be necessary?

**Reviewer Confidence:**

2: The reviewer is willing to defend the evaluation, but it is likely that the reviewer did not understand parts of the paper

**Scope:**

3: The work is somewhat relevant to the Web and to the track, and is of narrow interest to a sub-community

---

### Official Review · Reviewer_pkob · 2024-11-29

**Novelty:** 4
**Technical Quality:** 5

**Review:**

### Summary

This paper conducts a large-scale study of the security issues of docker images on Docker Hub. It proposed a framework, DITector, for the security analysis of Docker images. They collected information from over 12 million repos to analyze. Their analysis reveals five security issues, including software vulnerabilities, secret leaks, misconfigurations, malicious files, and sensitive parameters. Additionally, they considered the threats that might potentially be propagated through the reuse relationships between images, which was not fully studied by existing works. Their results show that security issues are widespread across the Docker image ecosystem and call for more studies to deal with the threats.

### Pros

1. Quality
   The paper has an ok quality in general. The objective of its study to analyze the security issues of Docker images is indeed critical, considering the wide usage of Docker. Their large-scale study and insight into dependencies between images that propagate security threats are quite impressive. Their analysis results about image distributions, five main security threats, and their propagation are also quite interesting.

2. Clarity
   The paper is generally easy to follow.

3. Originality
   The paper has an insightful observation that the Docker image ecosystem lacks large-scale analysis on security issues and threats propagation between the dependency of images (high-dependency-weight images). In terms of motivation and analysis, it is quite contributive.

4. Significance

   The paper provides significant insights into the Docker ecosystem and community through large-scale analysis. By describing five critical types of security threats and their propagation through image dependencies, it reveals that security issues are varied and widespread among Docker images.

### Cons

1. In terms of the detection approaches, the paper doesn't introduce many novel methods. They are mainly based on existing or third-party works, including Trufflehog for security leakage, Anchore for software vulnerability, and an antivirus engine from the corporate partner for malicious file detection. For misconfiguration detection, the paper manually reviews four popular containerized database applications. These detection methods do not present so much novelty.
2. Regarding software vulnerabilities in images, it selects 1000 high-pull-count images and 2000 high-dependency-weight images. It would be better to exploit a larger proportion of the dataset to better represent the whole.
3. It would be better to include more research directions in the discussion parts. The paper is good at summarizing the existing threats, but the discussion part is a bit limited. Besides giving suggestions to the users and developers, more discussion on the research directions to further protect Docker users from threats would be expected, especially given that there are many existing studies working on the security issues of Docker images.

**Questions:**

1. The sampling of the software vulnerabilities in images only contains a small portion of the entire dataset. Have you considered enlarging the scope to better represent the whole ecosystem?

2. Any specific suggestions about the focus of further studies on Docker security?

**Reviewer Confidence:**

3: The reviewer is confident but not certain that the evaluation is correct

**Scope:**

4: The work is relevant to the Web and to the track, and is of broad interest to the community

---

### Official Review · Reviewer_QfKU · 2024-12-01

**Novelty:** 6
**Technical Quality:** 6

**Review:**

The paper demonstrates high technical quality, employing systematic methodologies such as large-scale data collection, dependency graph modeling, and security analysis using a comprehensive framework. It integrates advanced analytical tools and presents findings with clear, quantitative metrics. It is well structured, with a logical flow from motivation to methodology, analysis, and results. The study is innovative in its large-scale approach to analyzing Docker image security, focusing on high-dependency-weight images and their role in threat propagation. It is impactful for understanding the Docker image ecosystem.

Cons: Focuses primarily on Docker Hub; other registries and container platforms are not considered. Reliance on existing tools like TruffleHog and Anchore may limit the scope of threat detection. Both of these were acknowledged in the limitation section of the paper.

**Questions:**

1. What were the specific limitations of tools like TruffleHog and Anchore in detecting threats? Did you consider developing custom detection mechanisms to address these limitations?

2. Were any biases introduced by focusing on high-pull-count and high-dependency-weight images, and how might these affect your conclusions?

3. Given the study’s focus on Docker Hub, how applicable are your findings to other registries or alternative container ecosystems (e.g., Quay.io, AWS Elastic Container Registry)?

**Reviewer Confidence:**

3: The reviewer is confident but not certain that the evaluation is correct

**Scope:**

4: The work is relevant to the Web and to the track, and is of broad interest to the community